# Zinc Complexes of Fluorosubstituted *N*-[2-(Phenyliminomethyl)phenyl]-4-methylbenzenesulfamides: Synthesis, Structure, Luminescent Properties, and Biological Activity

**DOI:** 10.3390/ma17020438

**Published:** 2024-01-17

**Authors:** Anatolii S. Burlov, Valery G. Vlasenko, Maxim S. Milutka, Yurii V. Koshchienko, Vladimir A. Lazarenko, Alexander L. Trigub, Alexandra A. Kolodina, Alexander A. Zubenko, Elena V. Braga, Alexey N. Gusev, Wolfgang Linert

**Affiliations:** 1Institute of Physical and Organic Chemistry, Southern Federal University, 344090 Rostov-on-Don, Russia; anatoly.burlov@yandex.ru (A.S.B.); milutka@sfedu.ru (M.S.M.); yukoshch@ipoc.sfedu.ru (Y.V.K.); akolodina@sfedu.ru (A.A.K.); 2Institute of Physics, Southern Federal University, 344090 Rostov-on-Don, Russia; v_vlasenko@rambler.ru; 3National Research Centre “Kurchatov Institute”, 123182 Moscow, Russia; lazarenko_va@nrcki.ru (V.A.L.); alexander.trigub@gmail.com (A.L.T.); 4North-Caucasian Zonal Scientific Research Veterinary Institute, Branch of the Federal State Budget Scientific Institution “Federal Rostov Agricultural Research Centre”, 344006 Rostov-on-Don, Russia; alexsandrzubenko@yandex.ru; 5General Chemistry Department, Crimean Federal University V.I. Vernadsky, 295007 Simferopol, Russia; braga.yelena@ya.ru; 6Institute of Applied Physics, Vienna University of Technology, Wiedner Hauptstraße 8-10, 1040 Vienna, Austria; wolfgang.linert@tuwien.ac.at

**Keywords:** azomethines, zinc(II) complexes, photoluminescence, electroluminescence, biological activity

## Abstract

Mono-, di-, and trifluorophenyl substituted in different positions of amine fragments bis [2-[[(*E*)-((fluorophenyl)iminomethyl]-*N*-(*p*-tolylsulfonyl)anilino]zinc(II) complexes were synthesized. Their crystal structure, photo- and electroluminescent properties, and protistocidal, fungistatic, and antibacterial activities were studied. It has been shown that the introduction of fluorine atoms and an increase in their number in the ligand structure of the resulting metal complexes promote the luminescence quantum yields and values of performance and brightness in EL cells compared to their previously studied chlorine-substituted analogs.

## 1. Introduction

The high demand for electroluminescent materials emitting in the range of 400–450 nm, i.e., blue emitters, is due to the fact that they are the main components of red–green–blue full-color displays and key electroluminescent components in the creation of white emitted by a blue and orange color combination [1]. Despite the fact that the number of luminophores emitting in the blue region is not inferior to the green and red phosphors, they significantly lose them in terms of stability in OLED (organic phosphors) and cost (iridium and osmium complexes). Therefore, the synthesis of new low-cost coordination compounds showing stable photoluminescence and electroluminescence in the blue region of the spectrum is still an important and urgent task.

The introduction of halogen atoms into ligand molecules, as previously shown, leads to an increase in the solubility and quantum yields (QY) of the photoluminescence (PL) and electroluminescence (EL) of their lanthanide(III) complexes [2,3,4,5,6]. A similar trend is characteristic of zinc(II) complexes based on the azomethine ligands 2-hydroxy- and 2-(*N*-tosylamino)benzaldehydes. The replacement of one or more C-H bonds by C-Cl and C-F bonds in the amine or aldehyde moieties of azomethines also leads to an increase in the PL QYs of the coordination compounds due to the quenching of vibrations [7,8,9]. The installation of the electron-withdrawing fluorine atoms helps to stabilize HOMO to widen the energy gap of the materials, thus realizing the blue emitter. It was found that zinc complexes with halogen-substituted azomethine ligands had photoluminescence quantum yields 2–4 times higher compared to unsubstituted ligands, both in solutions and in the solid state [10,11].

It has also been shown that, in azomethine metal complexes, the solubility and luminescence QYs increase significantly when the atomic number of the halogen substituent decreases and/or their number in the ligand increases. In this regard, the greatest effect is observed for fluorosubstituted azomethines and their metal complexes. In continuation of works on synthesis and the overall studies of PL and EL properties of the azomethine compounds of 2-hydroxybenzaldehydes and 2-*N*-tosylaminobenzaldehydes, here, we report the synthesis and comparative structural and photophysical studies of zinc complexes with fluoro-substituted azomethines and evaluate the impact of fluorine atoms’ position on their luminescent properties. We have also performed some preliminary studies by using them as emitters for the fabrication of electroluminescent devices.

## 2. Materials and Methods

Commercially available starting materials (Alfa Aesar, Ward Hill, MA, USA) were used as purchased: Zinc acetate dihydrate (CAS# 5970-45-6), 2-fluoroaniline (CAS# 348-54-9), 4-fluoroaniline (CAS# 371-40-4), 3,4-difluoroaniline (CAS# 3863-11-4), 2, 4-difluoroaniline (CAS No. 367-25-9), 2,5-difluoroaniline (CAS No. 367-30-6), 2,6-difluoroaniline (CAS No. 5509-65-9), 3,5-difluoroaniline (CAS No. 372-39-4), and 2,4,6-trifluoroaniline (CAS No. 363-81-5).

The C, H, and N elemental analyses were carried out on a «EuroEA-3000» (EuroVector, Milan, Italy) analyzer. The amount of the metal was determined by the gravimetric method. The IR spectra of the obtained complexes were recorded on a Varian 3100-FTIR (Varian, Australia) Excalibur instrument in the range 4000–400 cm^−1^ by the method of disturbed total internal reflection. The ^1^H NMR spectra were obtained on a Varian Unity-300 (Varian, Australia) instrument (300 MHz) in DMSO-*d*_6_.

The X-ray Zn K absorption edges of zinc complexes were obtained in the transmission mode at the Structural Materials Science station at the Kurchatov Synchrotron Center (Moscow, Russia) by the protocol described yearly [12]. The exact values of the nearest environment parameters of the zinc were determined by the IFFEFIT software package (version 1.2.11) [13,14]. More detailed information about the X-ray absorption experiment and EXAFS analysis can be found in the Appendix A.

For the single crystals of the free ligands **1d**, **1f**, and zinc(II) complexes **2d**, **2h**, and **2f**, X-ray diffraction data were collected at the ‘Belok’ beamline of the Kurchatov Synchrotron Radiation Source (NRC Kurchatov Institute, Moscow, Russia) [15]. All data were collected at 100 K. The data were indexed and integrated by the XDS and XSCALE software suites (version 30 June 2023) [16]. The structures were solved by direct methods (intrinsic phasing) with SHELXT [17]. The structural models were investigated by Olex2 software (Olex2-1.5) [18] and refined by SHELXL [19] by a full-matrix least-squares method on F^2^ with anisotropic displacements for all non-hydrogen atoms. Hydrogen atoms involved in H-bonding were refined isotropically. H-bonding-silent hydrogen atoms were placed into calculated positions and refined within the riding model with fixed isotropic displacement parameters.

The crystallographic parameters and the refinement statistics for **1d**, **1f**, **2d**, **2h**, and **2f** are given in Appendix A (Appendix A). Crystallographic data for these compounds have been deposited with the Cambridge Crystallographic Data Center, CCDC 2299391 (**1d**), 2299392 (**1f**), 2299394 (**2d**), 2299393 (**2h**), and 2299397 (**2f**), and can be obtained free of charge from the Cambridge Crystallographic Data Centre via www.ccdc.cam.ac.uk/data_request/cif (accessed on 12 December 2023).

The luminescence of complexes and azomethines was carried out for both solutions and solid samples. The spectra were recorded on a FluoroMax-4 spectrofluorimeter (HORIBA Scientific, Kyoto, Japan). Quantum yields of emission were determined by the absolute method using an integrating sphere. Lifetime measurements were performed on a Horiba Fluorocube instrument (HORIBA Scientific, Kyoto, Japan) by time-correlated single-photon counting using a 365 nm LED excitation source.

Fabrication of the OLED was performed according to the methodology described earlier [10]: using “AUTO 306” equipment by “BOC EDWARDS” (Crawley, UK) for the thermal deposition of layers of sand quartz and detector SQM 160 (INFICON GmbH, New York, NY, USA) for the control of evaporation speed and the thickness of the deposited layers.

The voltage-current and luminance measurements of the obtained OLED structures were studied on a measuring complex consisting of a voltage analyzer source (Keithley 237, KEITHLEY, Cleveland, OH, USA) and a fiber spectrometer (AvaSpec-ULS-2048 × 64, Avantes BV, Apeldoorn, The Netherlands).

A detailed description of the methodology for studying the biological activity of new substances is given in our own previous works [20,21] and the Appendix A.

### 2.1. General Procedure for the Synthesis of Azomethines ***1a**–**h***

A solution of 5 mmol of fluorosubstituted aniline in 3 mL of glacial acetic acid was added to a solution of 1.38 g (5 mmol) 2-(*N*-tosylamino)benzaldehyde [22] in 3 mL of glacial acetic acid; then, the reaction mixture was refluxed for 2 h after cooling to r. t. Six mL of EtOH were added. The precipitate was filtered off, recrystallized from acetic acid, and dried in a vacuum tube at 100 °C.

N-[2-[(*E*)-(2-Fluorophenyl)iminomethyl]phenyl]-4-methyl-benzenesulfonamide (**1a**) was prepared from 1.38 g (5 mmol) of 2-(*N*-tosylamino)benzaldehyde and 0.56 g (5 mmol) 2-fluoroaniline. Yield 1.60 g (87%), orange powder, m.p. 133–134 °C (AcOH). IR spectrum (vas. oil), ν, cm^−1^: 3285 (NH), 1624 s (CH=N), 1599, 1574, 1488, 1456, 1416, 1377, 1342 s (as SO_2_), 1308, 1288, 1248, 1161 s (s SO_2_), 1120, 1091, 1047, 971, 946, 879, 840, 814, 798, 759, 661, 619, 560. ^1^HNMR (300 MHz, DMSO-*d*_6_) δ: 2.28 (s, 3H, CH_3_), 7.20 (tt, 1H, 3*J* = 7.4 Hz, 4*J* = 1.2 Hz, CAr-H), 7.30–7.38 (m, 5H, CAr-H), 7.43–7.53 (m, 3H, CAr-H), 7. 68 (d, 2H, 3*J* = 8.1 Hz, CAr-H), 7.76 (dd, 1H, 3*J* = 7.8 Hz, 4*J* = 1.2 Hz, CAr-H), 8.83 (s, 1H, CH=N), 12.46 (s, 1H, NH). Found, %: C 65.24; N 4.71; N 7.65. C_20_H_17_FN_2_O_2_S. Calculated, %: C 65.20; N 4.65; N 7.60.*N*-[2-[(*E*)-(4-Fluorophenyl)iminomethyl]phenyl]-4-methyl-benzenesulfonamide (**1b**) was prepared from 1.38 g (5 mmol) of 2-(*N*-tosylamino)benzaldehyde and 0.56 g (5 mmol) 4-fluoroaniline. Yield 1.55 g (85%), orange powder, m.p. 146–147 °C (AcOH). IR spectrum (vas. oil), ν, cm^−1^: 3287, 3210 (NH), 1621 s (CH=N), 1596, 1573, 1497, 1463, 1455, 1427, 1402, 1378, 1338 s (as SO_2_), 1309, 1292, 1243, 1228, 1167 s (s SO_2_), 1155, 1117, 1090, 1047, 1019, 950, 884, 853, 833, 799, 776, 751, 728, 707, 662, 637, 616, 592, 568. ^1^H NMR (300 MHz, DMSO-*d*_6_) δ: 2.28 (s, 3H, CH_3_), 7.17–7.22 (m, 1H, CAr-H), 7.29–7.41 (m, 6H, CAr-H), 7.44 (d, 2H, 4*J* = 3.6 Hz, CAr-H), 7.67 (d, 2H, 3*J* = 8.4 Hz, CAr-H), 7.74 (d, 1H, 3*J* = 7.5 Hz, CAr-H), 8.74 (s, 1H, CH=N), 12.43 (s, 1H, NH). Found, %: C 65.10; N 4.75; N 7.54. C_20_H_17_FN_2_O_2_S. Calculated, %: C 65.20; N 4.65; N 7.60.*N*-[2-[[(*E*)-(2,4-Difluorophenyl)iminomethyl]phenyl]phenyl]-4-methyl-benzenesulfonamide (**1c**) was prepared from 1.38 g (5 mmol) of 2-(*N*-tosylamino)benzaldehyde and 0.65 g (5 mmol) 2,4-difluoroaniline. The yield is 1.69 g (88%), an orange powder, m.p. 176–177 °C (AcOH). IR spectrum (vas. oil), ν, cm^−1^: 3057 (NH), 1626 s (CH=N), 1597, 1574, 1540, 1495, 1463, 1403, 1377, 1339 s (as SO_2_), 1291, 1267, 1229, 1168 s (s SO_2_), 1156, 1142, 1116, 1092, 1047, 1047, 966, 944, 944, 875, 839, 823, 809, 751, 728, 708, 663, 638, 619, 567. ^1^H NMR (300 MHz, DMSO-*d*_6_) δ: 2.29 (s, 3H, CH_3_), 7.18–7.24 (m, 2H, CAr-H), 7.31 (d, 2H, 3*J* = 8.1 Hz, CAr-H), 7.40–7.58 (m, 4H, CAr-H), 7.66 (d, 2H, 3*J* = 8.1 Hz, CAr-H), 7.75 (d, 1H, 3*J* = 7.8 Hz, CAr-H), 8.82 (s, 1H, CH=N), 12.36 (s, 1H, NH). Found, %: C 62.09; N 4.28; N 7.32. C_20_H_16_F_2_N_2_O_2_S. Calculated, %: C 62.16; N 4.17; N 7.25.*N*-[2-[[(*E*)-(2,5-Difluorophenyl)iminomethyl]phenyl]phenyl]-4-methyl-benzenesulfonamide (**1d**) was prepared from 1.38 g (5 mmol) of 2-(*N*-tosylamino)benzaldehyde and 0.65 g (5 mmol) 2,5-difluoroaniline. The yield is 1.53 g (79%), orange powder, m.p. 170–171 °C (AcOH). IR spectrum (vas. oil), ν, cm^−1^: 3254 (NH), 1629 m (CH=N), 1603, 1573, 1495, 1463, 1418, 1404, 1378, 1342 s (as SO_2_), 1309, 1289, 1273, 1251, 1224, 1166, 1161 s (s SO_2_), 1143, 1119, 1091, 1049, 1022, 971, 941, 873, 855, 842, 824, 814, 805, 761, 748, 724, 706, 662, 620, 607, 590, 563. ^1^H NMR (300 MHz, DMSO-*d*_6_) δ: 2.28 (s, 3H, CH_3_), 7.19–7.25 (m, 2H, CAr-H), 7.31 (d, 2H, 3*J* = 8.4 Hz, CAr-H), 7.36–7.43 (m, 2H, CAr-H), 7.48–7.50 (m, 2H, CAr-H), 7.67 (d, 2H, 3*J* = 8.4 Hz, CAr-H), 7.76 (d, 1H, 3*J* = 7.8 Hz, CAr-H), 8.82 (s, 1H, CH=N), 12.22 (s, 1H, NH). Found, %: C 62.07; N 4.25; N 7.36. C_20_H_16_F_2_N_2_O_2_S. Calculated, %: C 62.16; N 4.17; N 7.25.*N*-[2-[(*E*)-(2,6-Difluorophenyl)iminomethyl]phenyl]-4-methyl-benzenesulfonamide (**1e**) was prepared from 1.38 g (5 mmol) of 2-(*N*-tosylamino)benzaldehyde and 0.65 g (5 mmol) of 2,6-difluoroaniline. The yield is 1.78 g (92%), orange powder, m.p. 153–154 °C (AcOH). IR spectrum (vas. oil), ν, cm^−1^: 3287, 3210, 3127 (NH), 1668, 1625 s (CH=N), 1600, 1572, 1495, 1479, 1470, 1407, 1379, 1343 s (as SO_2_), 1311, 1285, 1244, 1221, 1121, 1184, 1171 s (as SO_2_), 1157, 1117, 1091, 1047, 1013, 973, 937, 837, 871, 847, 817, 798, 779, 755, 737, 719, 661, 627, 565. ^1^HNMR (300 MHz, DMSO-*d*_6_) δ: 2.30 (s, 3H, CH_3_), 7.21–7.33 (m, 6H, CAr-H), 7.49 (dd, 2H, 3*J* = 7.5 Hz, 4*J* = 1.5 Hz, CAr-H), 7.66 (d, 2H, 3*J* = 8.4 Hz, CAr-H), 7.77 (d, 1H, 3*J* = 7.5 Hz, CAr-H), 8.87 (s, 1H, CH=N), 11.98 (s, 1H, NH). Found, %: C 62.08; N 4.24; N 7.32. C_20_H_16_F_2_N_2_O_2_S. Calculated, %: C 62.16; N 4.17; N 7.25.*N*-[2-[[[(*E*)-(3,4-Difluorophenyl)iminomethyl]phenyl]phenyl]-4-methyl-benzenesulfonamide (**1f**) was prepared from 1.38 g (5 mmol) of 2-(*N*-tosylamino)benzaldehyde and 0.65 g (5 mmol) 3,4-difluoroaniline. The yield is 1.69 g (88%), an orange powder, m.p. 174–175 °C (AcOH). IR spectrum (vas. oil), ν, cm^−1^: 3058 (NH), 1627 m (CH=N), 1600, 1573, 1549, 1514, 1456, 1418, 1399, 1377, 1338 s (as SO_2_), 1310, 1292, 1283, 1259, 1217, 1203, 1169 s (as SO_2_), 1157, 1140, 1120, 1105, 1090, 1047, 1019, 957, 947, 879, 864, 851, 822, 806, 786, 758, 739, 723, 707, 693, 662, 638, 618, 594, 567. ^1^HNMR (300 MHz, DMSO-*d*_6_) δ: 2.28 (s, 3H, CH3), 7.15–7.24 (m, 2H, CAr-H), 7.31 (d, 2H, 3*J* = 8.1 Hz, CAr-H), 7.40–7.59 (m, 4H, CAr-H), 7.67 (d, 2H, 3*J* = 8.1 Hz, CAr-H) 7.76 (dd, 1H, 3*J* = 7.6 Hz, 4*J* = 1.4 Hz, CAr-H), 8.71 (s, 1H, CH=N), 12.06 (s, 1H, NH). Found, %: C 62.10; N 4.28; N 7.32. C_20_H_16_F_2_N_2_O_2_S. Calculated, %: C 62.16; N 4.17; N 7.25.*N*-[2-[(*E*)-(3,5-Difluorophenyl)iminomethyl]phenyl]-4-methyl-benzenesulfonamide (**1g**) was prepared from 1.38 g (5 mmol) of 2-(*N*-tosylamino)benzaldehyde and 0.65 g (5 mmol) of 3,5-difluoroaniline. The yield is 1.60 g (83%), orange powder, m.p. 146–147 °C (AcOH). IR spectrum (vas. oil), ν, cm^−1^: 3130, 3088 (NH), 1600 s (CH=N), 1572, 1522, 1503, 1495, 1464, 1454, 1415, 1378, 1346 s (as SO_2_), 1323, 1307, 1290, 1251, 1226, 1208, 1187, 1169 s (s SO_2_), 1156, 1131, 1119, 1090, 1048, 1020, 1008, 997, 986, 943, 869, 852, 841, 818, 804, 790, 762, 737, 724, 706, 660, 624, 583, 565. ^1^HNMR (300 MHz, DMSO-*d*_6_) δ: 2. 29 (s, 3H, CH_3_), 7.03 (dd, 2H, 3*J* = 8.5 Hz, 4*J* = 2.2 Hz, CAr-H), 7.15–7.26 (m, 2H, CAr-H), 7.31 (d, 2H, 3*J* = 8.4 Hz, CAr-H), 7.39 (d, 1H, 3*J* = 8.1 Hz, CAr-H), 7.48 (tt, 1H, 3*J* = 8.4 Hz, 4*J* = 1.5 Hz, CAr-H), 7.66 (d, 2H, 3*J* = 8.4 Hz, CAr-H), 7.79 (dd, 1H, 3*J* = 7.8 Hz, 4*J* = 1.5 Hz, CAr-H), 8.68 (s, 1H, CH=N), 11.75 (s, 1H, NH). Found, %: C 62.07; N 4.28; N 7.33 C_20_H_16_F_2_N_2_O_2_S. Calculated, %: C 62.16; N 4.17; N 7.25.4-Methyl-*N*-[2-[(*E)*-(2,4,6-trifluorophenyl)iminomethyl]phenyl]-benzenesulfonamide (**1h**) was prepared from 1.38 g (5 mmol) of 2-(*N*-tosylamino)benzaldehyde and 0.74 g (5 mmol) 2,4,6-trifluoroaniline. The yield is 1.72 g (85%), white powder, m.p. 170–171 °C (AcOH). IR spectrum (vas. oil), ν, cm^−1^: 3129, 3056 (NH), 1634 s (CH=N), 1613, 1595, 1572, 1486, 1461, 1403, 1379, 1341 s (as SO_2_), 1310, 1286, 1237, 1223, 1186, 1171 s (s SO_2_), 1155, 1121, 1091, 1048, 1020, 1000, 974, 936, 869, 850, 842, 821, 799, 756, 707, 664, 630, 609, 567. ^1^HNMR (300 MHz, DMSO-*d*_6_) δ: 2.30 (s, 3H, CH_3_), 7.22 (tt, 1H, 3*J* = 7.4 Hz, 4*J* = 1.5 Hz, CAr-H), 7.32 (d, 2H, 3*J* = 8.1 Hz, CAr-H), 7.37–7.50 (m, 4H, CAr-H), 7.65 (d, 2H, 3*J* = 8.4 Hz, CAr-H), 7.78 (dd, 1H, 3*J* = 8.7 Hz, 4*J* = 1.2 Hz, CAr-H), 8.86 (s, 1H, CH=N), 11.89 (s, 1H, NH). Found, %: C 59.32; N 3.79; N 7.01. C_20_H_15_F_3_N_2_O_2_S. Calculated, %: C 59.40; N 3.74; N 6.93.

### 2.2. General Procedure for the Synthesis of Complexes ***2a**–**h***

A solution of 0.22 g (1 mmol) of Zn(CH_3_COO)_2_·2H_2_O in 5 mL of methanol was added to a hot solution of 2 mmol azomethine **1a**–**h** in 30 mL of a mixture of methanol and chloroform (1:1). Further, 0.08 g (2 mmol) of NaOH in 5 mL of methanol were added dropwise. The single crystals of zinc(II) complexes were grown by slow evaporation at the room temperature of their solutions in a mixture of methylene chloride and methanol (1:2).

Bis[2-[[(*E*)-(2-fluorophenyl)iminomethyl]-*N*-(*p*-tolylsulfonyl)anilino]zinc(II) (**2a**) was obtained from 0.74 g (2 mmol) of azomethine **1a**. The yield is 0.60 g (75%), yellow powder, m.p. > 300 °C. IR spectrum (vas. oil), ν, cm^−1^: 1615 s (CH=N), 1605, 1555, 1480, 1461, 1403, 1377, 1300 s (as SO_2_), 1284, 1266, 1207, 1172, 1139 s (s SO_2_), 1104, 1081, 1056, 954, 932, 901, 861, 846, 833, 812, 787, 756, 722, 666, 617, 594, 567. ^1^H NMR (300 MHz, DMSO-*d*_6_) δ: 2.28 (s, 3H, CH_3_), 6.90 (t, 1H, 3*J* = 6.3 Hz, CAr-H), 7.10–7.33 (m, 7H, CAr-H), 7.39 (t, 1H, 3*J* = 7.8 Hz, CAr-H), 7.65–7.73 (m, 3H, CAr-H), 8.74 (s, 1H, CH=N). Found, %: C 59.96; H 4.15; N 7.09; Zn 8.10. C_40_H_32_F_2_N_4_O_4_S_2_Zn. Calculated, %: C 60.04; H 4.03; N 7.00; Zn 8.17.Bis[2-[[(*E*)-(4-fluorophenyl)iminomethyl]-*N*-(*p*-tolylsulfonyl)anilino]zinc(II) (**2b**) was prepared from 0.74 g (2 mmol) of azomethine **1b**. The yield is 0.58 g (73%), yellow powder, m.p. > 300 °C. IR spectrum (vas. oil), ν, cm^−1^: 1613 s (CH=N), 1598, 1556, 1504, 1480, 1464, 1446, 1399, 1377, 1297 s (as SO_2_), 1287, 1258, 1233, 1176, 1138 s (s SO_2_), 1081, 1056, 1022, 955, 905, 902, 880, 856, 833, 813, 780, 758, 713, 666, 645, 619, 585, 560. ^1^H NMR (300 MHz, DMSO-*d*_6_) δ: 2.30 (s, 3H, CH3), 6.92–6.96 (m, 1H, CAr-H), 7.15–7.22 (m, 4H, CAr-H), 7.32–7.36 (m, 4H, CAr-H), 7.65 (d, 2H, 3*J* = 8.1 Hz, CAr-H), 7.74 (d, 1H, 3*J* = 7.8 Hz, CAr-H), 8.74 (s, 1H, CH=N). Found, %: C 60.00; H 4.14; N 6.91; Zn 8.10. C_40_H_32_F_2_N_4_O_4_S_2_Zn. Calculated, %: C 60.04; H 4.03; N 7.00; Zn 8.17.Bis[2-[[(*E*)-(2,4-difluorophenyl)iminomethyl]-*N*-(*p*-tolylsulfonyl)anilino] zinc(II) (**2c**) was prepared from 0.77 g (2 mmol) of azomethine **1c**. The yield is 0.60 g (72%), yellow powder, m.p. > 300 °C. IR spectrum (vas. oil), ν, cm^−1^: 1613 s (CH=N), 1556, 1503, 1482, 1462, 1445, 1402, 1377, 1301 s (as SO_2_), 1283, 1264, 1219, 1172, 1139 (s SO_2_), 1097, 1081, 1022, 1007, 970, 954, 933, 933, 898, 850, 842, 813, 757, 738, 713, 665, 646, 611, 578. ^1^HNMR (300 MHz, DMSO-*d*_6_) δ: 2.27 (s, 3H, CH_3_), 6.91 (t, 1H, 3*J* = 7.1 Hz, CAr-H), 7.04 (t, 1H, 3*J* = 7.5 Hz, CAr-H), 7.17 (d, 2H, 3*J* = 7.8 Hz, CAr-H),7.21–7.30 (m, 3H, CAr-H), 7.43 (q, 1H, 3*J* = 6.3 Hz, CAr-H), 7.66 (d, 1H, 3*J* = 7.5 Hz, CAr-H), 7.77 (d, 2H, 3*J* = 7.8 Hz, CAr-H), 8.74 (s, 1H, CH=N). Found, %: C 57.40; H 3.69; N 6.75; Zn 7.72. C_40_H_30_F_4_N_4_O_4_S_2_Zn. Calculated, %: C 57.45; H 3.62; N 6.70; Zn 7.82.Bis[2-[[(*E*)-(2,5-difluorophenyl)iminomethyl]-*N*-(*p*-tolylsulfonyl)anilino]zinc(II) (**2d**) was prepared from 0.77 g (2 mmol) of azomethine **1d**. The yield is 0.57 g (68%), yellow powder, m.p. 289–290 °C. IR spectrum (vas. oil), ν, cm^−1^: 1603 s (CH=N), 1553, 1525, 1495, 1481, 1463, 1454, 1434, 1409, 1377, 1338, 1302 s (as SO_2_), 1290, 1263, 1207, 1193, 1148, 1134 s (s SO_2_), 1100, 1081, 1061, 1021, 1011, 969, 942, 895, 895, 874, 842, 822, 814, 786, 758, 734, 711, 664, 643, 610, 574, 552. ^1^HNMR (300 MHz, DMSO-*d*_6_) δ: 2.27 (s, 3H, CH3), 6.92 (t, 1H, 3*J* = 7.2 Hz, CAr-H), 7.09–7.34 (m, 7H, CAr-H), 7. 67 (d, 1H, 3*J* = 7.8 Hz, CAr-H), 7.77 (d, 2H, 3*J* = 8.1 Hz, CAr-H), 8.77 (s, 1H, CH=N). Found, %: C 57.39; H 3.68; N 6.74; Zn 7.89. C_40_H_30_F_4_N_4_O_4_S_2_Zn. Calculated, %: C 57.45; H 3.62; N 6.70; Zn 7.82.Bis[2-[[(*E*)-(2,6-difluorophenyl)iminomethyl]-*N*-(*p*-tolylsulfonyl)anilino]zinc(II) (**2e**) was prepared from 0.77 g (2 mmol) of azomethine **1e**. The yield is 0.64 g (77%), yellow powder, m.p. > 300 °C. IR spectrum (vas. oil), ν, cm^−1^: 1615 s (CH=N), 1603, 1552, 1476, 1441, 1413, 1377, 1301 s (as SO_2_), 1287, 1263, 1242, 1182, 1172, 1140 s (s SO_2_), 1081, 1055, 1017, 982, 946, 896, 849, 827, 811, 772, 757, 738, 713, 663, 642, 617, 598, 578. ^1^H NMR (300 MHz, DMSO-*d*_6_) δ: 2.24 (s, 3H, CH3), 6.87–7.25 (m, 8H, CAr-H), 7.64–7.86 (m, 3H, CAr-H), 8.84 (s, 1H, CH=N). Found, %: C 57.40; H 3.69; N 6.75; Zn 7.76. C_40_H_30_F_4_N_4_O_4_S_2_Zn. Calculated, %: C 57.45; H 3.62; N 6.70; Zn 7.82.Bis[2-[[(*E*)-(3,4-difluorophenyl)iminomethyl]-*N*-(*p*-tolylsulfonyl)anilino]zinc(II) (**2f**) was prepared from 0.77 g (2 mmol) of azomethine **1f**. The yield is 0.59 g (70%), yellow powder, m.p. > 300 °C. IR spectrum (vas. oil), ν, cm^−1^: 1599 s (CH=N), 1555, 1510, 1479, 1464, 1448, 1396, 1377, 1300 s (as SO_2_), 1263, 1199, 1177, 1135 s (s SO_2_), 1112, 1081, 1057, 1023, 974, 947, 895, 864, 839, 814, 788, 756, 723, 708, 666, 635, 620, 578. ^1^HNMR (300 MHz, DMSO-*d*_6_) δ: 2.30 (s, 3H, CH_3_), 6.96 (t, 1H, 3J = 7.2 Hz, CAr-H), 7.12–7.20 (m, 3H, CAr-H), 7.29–7.48 (m, 4H, CAr-H), 7.69 (d, 2H, 3*J* = 8.4 Hz, CAr-H), 7.74 (d, 1H, 4*J* = 1.5 Hz, CAr-H), 8.73 (s, 1H, CH=N). Found, %: C 57.40; H 3.68; N 6.95; Zn 7.88. C_40_H_30_F_4_N_4_O_4_S_2_Zn. Calculated, %: C 57.45; H 3.62; N 6.70; Zn 7.82.Bis[2-[[(*E*)-(3,5-difluorophenyl)iminomethyl]-*N*-(*p*-tolylsulfonyl)anilino]zinc(II) (**2g**) was prepared from 0.77 g (2 mmol) of azomethine **1g**. The yield is 0.66 g (79%), yellow powder, m.p. > 300 °C. IR spectrum (vas. oil), ν, cm^−1^: 1601 s (CH=N), 1555, 1519, 1486, 1468, 1456, 1420, 1377, 1326, 1299 s (as SO_2_), 1263, 1220, 1207, 1186, 1152, 1131 s (s SO_2_), 1081, 1062, 1041, 1016, 988, 968, 944, 894, 867, 846, 835, 822, 756, 721, 710, 679, 663, 646, 624, 567. ^1^H NMR (300 MHz, DMSO-*d*_6_) δ: 2.30 (s, 3H, CH3), 6.95–7.02 (m, 3H, CAr-H), 7.17–7.30 (m, 4H, CAr-H), 7.37 (tt, 1H, 3J = 8. 4 Hz, 4*J* = 1.2 Hz, CAr-H), 7.71 (d, 2H, 3*J* = 8.1 Hz, CAr-H), 7.74 (s, 1H, CAr-H), 8.77 (s, 1H, CH=N). Found, %: C 57.40; H 3.72; N 6.78; Zn 7.88. C_40_H_30_F_4_N_4_O_4_S_2_Zn. Calculated, %: C 57.45; H 3.62; N 6.70; Zn 7.82.Bis[*N*-(*p*-tolylsulfonyl)-2-[(*E*)-(2,4,6-trifluorophenyl)iminomethyl]anilino]zinc(II) (**2h**) was prepared from 0.81 g (2 mmol) of azomethine **1h**. The yield is 0.67 g (77%), yellow powder, m.p. > 300 °C. IR spectrum (vas. oil), ν, cm^−1^: 1610 s (CH=N), 1556, 1501, 1479, 1454, 1441, 1406, 1377, 1358, 1302 s (as SO_2_), 1280, 1266, 1230, 1181, 1141 s (s SO_2_), 1119, 1084, 1047, 1021, 9921, 998, 954, 898, 898, 862, 836, 826, 767, 757, 730, 709, 667, 645, 610, 579, 554. ^1^HNMR (300 MHz, DMSO-*d*_6_) δ: 2.23 (s, 3H, CH_3_), 6.88 (t, 1H, 3*J* = 7.2 Hz, CAr-H), 7.16 (d, 4H, 3*J* = 8.1 Hz, CAr-H), 7.22–7.31 (m, 2H, CAr-H), 7.62 (d, 1H, 3J = 6.9 Hz, CAr-H), 7.90 (d, 2H, 3*J* = 7.8 Hz, CAr-H), 8.85 (s, 1H, CH=N). Found, %: C 55.01; H 3.29; N 6.47; Zn 7.38. C_40_H_28_F_6_N_4_O_4_S_2_Zn. Calculated, %: C 55.08; H 3.24; N 6.42; Zn 7.50.

## 3. Results

### 3.1. Synthesis and Spectroscopic Studies of Azomethines ***1a**–**h*** and Zinc(II) Complexes ***2a**–**h***

The synthesis of azomethines **1a**–**h** and zinc complexes **2a**–**h** on their basis is presented in Figure 1.

The IR spectra of azomethines **1a**–**h** show weakly intense νNH absorption bands in the region of 3058–3287 cm^−1^ and νCH=N absorption bands in the region of 1600–1634 cm^−1^, ν_as_SO_2_ 1338–1346 cm^−1^ and ν_s_SO_2_ 1161–1171 cm^−1^. The azomethines **1a**–**h**
^1^H NMR spectra contain the proton signals of NH groups at 11.75–12.46 ppm and the CH=N group’s proton signals at 8.68–8.87 ppm.

By the reaction of azomethine **1a**–**h** and zinc acetate dihydrate (molar ratio of ligand: zinc acetate is 2:1), we obtained the complexes **2a**–**h**. These complexes are yellow crystalline substances with an m.p. of >300 °C and soluble in methylene chloride, DMFA, and DMSO. The composition of the complexes is ZnL_2_ according to the elemental analysis data. The absorption bands of the νNH ligand disappear in the IR spectra of the complexes. The absorption bands of νCH=N undergo a shift to the long wavelength (low frequency) region by 8–28 cm^−1^ and the bands of νasSO_2_ by 38–47 cm^−1^ and νsSO_2_ by 22–28 cm^−1^. The formation of zinc complexes is also indicated by the disappearance of the NH group of ligands **1a**–**h** signal in the ^1^H NMR spectra. In addition, the signals of the CH=N group’s protons are shifted slightly to the strong field, which is typical for the formation of chelate structures [20,21].

### 3.2. Crystal Structures of ***1d**,**f***

The molecular structures of **1d** and **1f** are shown in Figure 1.

Crystal **1d**, unlike crystal **1f**, consists of two crystallographically independent molecules. The geometric parameters for all molecules (Table 1) are within the typical ranges observed for other Schiff base ligands with tosylamine fragments [23,24]. Each iminomethylphenyl fragment is planar, where the angles between the planes are about 2°. The angles between benzene cycles of the tosylamine fragments and the iminomethylphenyl fragments are close to orthogonal (82.89(6)° and 89.38(6)° for **1d**, 78.28(9)° for **1f**). The angles between the iminomethylphenyl fragment and the plane of the fluorine-substituted aniline fragment in **1d** are 19.63(5)° and 14.37(6)°, while, in **1f**, these angles are much smaller and equal to about 4.74(4)°. All t polyhedron’s angles are close to ideal tetrahedral geometry, except the angle O1=S1=O2 (119.7°). The S1=O1 and S1=O2 bond distances in azomethines **1d** and **1f** are very close and range from 1.5209(2) to 1.5278(1) Å. The S1-N1 bond lengths, 1.7267(1)–1.7366(2) Å, are very close to the single bond lengths (1.74 Å).

The molecular structures of each ligand **1d**,**f** contain two specific intramolecular interactions (Figure 2). One of them, an intramolecular hydrogen bond N–H…N, differs significantly for these two compounds. The H-bonds in the crystal of **1d** are formed as N2–H2…N1, with bond lengths 2.098 and 2.119 Å; while in **1f**, this bond is due to N1–H1…N2 and is much shorter than 1.999 Å (Table 2). The existence of such strong hydrogen bonds in the molecules of azomethines **1d**,**f** leads to the formation of an almost planar bicyclic iminomethylphenyl system, significantly increasing their structural rigidity [25]. The second specific interaction is the intramolecular hydrogen bond C9-H…O2=S1 contact with interatomic distances of 2.477 and 2.537 Å in **1d** and 2.564 Å in **1f**, respectively (Table 2).

The crystal packing of molecule **1d** is due to intermolecular hydrogen bonds between the oxygen atoms of the tosylamine group of the ligand of one of the molecules and the hydrogen atom of the fluorine-substituted aniline fragment, with the H-bond length 2.651 Å, and the methyl group of the tosylamine fragment, with the H-bond length 2.691 Å of neighboring molecules (Table 2). The result of such interactions is the formation of infinitely elongated chains along the [b] direction in the crystal. Crystal **1d** also exhibits π-stacking interactions between the rings of fluorine-substituted aniline fragments, with centroid distances of 3.964 Å, and the rings of iminomethylphenyl fragments, with centroid distances of 4.049 Å of neighboring molecules.

The crystal packing of molecule **1f** is determined by intermolecular hydrogen bonds between the oxygen atoms of the tosylamine group of the ligand and the tosylamine fragment of the neighboring molecule with an H-bond length of 2.685 Å. Owing to antiparallel π-stacking interactions of rings of iminomethylphenyl and fluorine-substituted aniline fragments, the stacks of molecules elongated in the [a] direction are formed (Figure 2).

### 3.3. X-ray Absorption Spectroscopy of Zinc(II) Complexes ***2a**–**h***

The XANES and EXAFS X-ray absorption spectroscopy of the Zn K absorption edges was used to characterize the local atomic environment of zinc ions in complexes **2a**–**h**. Figure 3 shows normalized XANES and the corresponding MFT (Modules of Fourier Transform) EXAFS for all zinc(II) complexes **2a**–**h**. It is known that the X-ray absorption edge depends both on the oxidized state of the metal ion and on the chemical environment, viz., effective charge, nature of ligands, coordination numbers, electronegativity of anions, and covalent character of the bonds surrounding the metal ion. The absorption edge and white-line position’s characteristics of the spectral features of the Zn K absorption edges for **2a**–**h** are similar (Figure 3a), indicating a similar environment of zinc ions in these compounds. In the XANES spectra of complexes **2a**–**h**, there is no pre-edge peak A due to the filled 3d shell of Zn(II). The energy positions of intense peak *C* (white line) correspond to the maximum of the X-ray absorption spectrum. And, the postedge peak *D* has some differences in the case of complexes **2e** and **2h**, where fluorine atoms are in the 2,6 positions of the aniline fragment of ligands.

The main characteristics of the coordination polyhedron for complexes **2a**–**h** were determined by EXAFS analysis. The EXAFS MFTs of these compounds are shown in Figure 3b. All the MFTs have a main peak at r = 1.51–1.53 Å, which corresponds to the photoelectron wave scattering by the nearest coordination sphere (CS) of the nitrogen atoms of the ligands. The MFT peaks at larger values of r > 2.5 Å are associated with the subsequent CSs containing different ligand atoms, mainly carbon atoms, as well as oxygen and sulfur tosylamine fragments of the ligands. It can be noted that, in the MFT of complexes **2e** and **2h** at r = 2.85–2.90 Å, there is a peak of high amplitude, which we interpreted as a manifestation of photoelectron scattering on fluorine atoms in the two and six positions of the aniline fragment of the ligands. The EXAFS model’s calculations show that the nearest environment of zinc ions in all **2a**–**h** complexes is similar and consists of four nitrogen atoms with average distances of Zn…N about 1.97–1.99 Å and 2.02–2.05 Å (Table 3). The obtained values of the Debye–Waller coefficients were about 0.0030 Å^2,^ which agrees with similar values for the analogous complexes determined earlier [10,11].

### 3.4. A Single-Crystal X-ray Diffraction of Zinc(II) Complexes ***2d**,**h**,**f***

Single-crystal X-ray diffraction analysis revealed that complexes **2d**,**h** crystallized in the monoclinic space group C2/c, whereas complex **2f** crystallized in the triclinic space group P-1, respectively. The summary of selected bond lengths and angles for the molecules in the complexes are shown in Table 4. The molecular geometries of the complexes **2d**,**h**,**f** were quite similar, as depicted in Figure 4, Figure 5 and Figure 6.

The compounds **2d**,**h** form centrosymmetric mononuclear molecules with two Schiff base ligands. In all three complexes, the zinc ions have an oxidation state of 2+ and a bicapped tetrahedral coordination environment “4 + 2” by four N atoms from the tosylamine and imine groups [Zn1-N2 1.9857(19) Å, Zn1-N1 2.0561(19) Å for **2d**, Zn1-N2 1.9938(11) Å, Zn1-N1 2.0396(11) Å for **2h**, and Zn1-N3,N4 1.9865(15) Å, 1.9905(15) Å, Zn1-N1,N2 2.0564(17), 2.0597(16) Å for **2h**] and additional weaker interactions with two O atoms from the sulfo groups [Zn1···O2,O2a 2.670 Å, and 2.719 Å for **2d**,**h**, and Zn1···O2 2.590 Å, and Zn1…O3 2.653 Å for **2f**]. The average values of the bond lengths coincide with the average values of such bonds from the CSD for similar complexes [20,21,26]. The bond lengths obtained by XRD show good agreement with the EXAFS data of the relative compounds. The bond angles N-Zn1-N in the coordination sphere of complexes **2d**,**h**,**f** vary in the interval from 90.81 to 149.28 deg., and, therefore, the zinc coordination polyhedron in these compounds can be described as a distorted tetrahedron. The angular structural parameter τ_4_ for the four coordinate complexes [27] was equal to 0.76 (**2d**), 0.67 (**2h**), and 0.73 (**2f**), which fits with a seesaw description.

The crystal packing of complex **2d** is enhanced by intermolecular hydrogen bonds C18-H18…F2, C16-H16…F1 of fluorine atoms of a fluorine-substituted aniline fragment and benzene cycles of tosylamine fragments, as well as between oxygen atoms of the sulfo group of one of the tosylamine fragments with the methyl group of the tosylamine fragment of neighboring molecule C20-H20B…O1 (Table 5).

In contrast to the crystalline packing of complex **2d**, the crystal lattice of complex **2h** contains a methanol solvate molecule that forms with the complex H-bonds molecule with iminomethylphenyl fragments C6-H7…O3, C9-H9…O3, and O3-H3…O2 with the sulfo group of the tosylamine fragment of the ligand. Intermolecular interactions C6-H6…F2, and the interactions between the two fluorine-substituted aniline fragments of the ligands, are also present (Table 5).

As in the crystal of complex **2h**, in the unit cell **2f**, there is a solvate molecule, methanol, which forms hydrogen bonds O5-H5A…F5 and C5-H5A…F4 with fluorine-substituted aniline fragments of ligands in the molecules of the complex. Fluorine of the (phenyl)iminomethyl fragments has two conformational positions with different occupancies, which were defined separately. In the case of conformation “part 2”, F1 is too close to the O5 atom of methanol (2.005 Å). Thus, the methanol molecule is present in the structure of **2f** only in the case of conformation “part 1”. Thus, the fractional chemical formula C_40_H_30_F_4_N_4_O_4_S_2_Zn, 0.632(CH_4_O) is a consequence of the fractional population of methanol. In addition, hydrogen bonds are formed between the oxygen atoms of the sulfo group with the phenyl ring of the aldehyde fragment and the phenyl ring of the tosylamine fragment of ligands.

The crystal structure of **2f** is strengthened by π-π interactions between benzene rings of the tosyl fragments of ligands with centroid–centroid distances of 3.779 Å and 3.821 Å and shift distances of 1.419 and 1.060 Å, respectively. Moreover, π–π interactions between rings of fluorine-substituted aniline fragments with centroid distances of 3.618 Å and a shift distance of 1.261 Å, increasing the stability of crystals, are present.

### 3.5. The Photoluminescent Properties

The luminescent properties of azomethine **1a**–**f** and Zn complexes **2a**–**f** were studied both in the solid state and a dichloromethane solution at room temperature. The data are depicted in Table 6. In the solid state, the parent ligands exhibit orange–red luminescence in the form of broad band luminescence with maxima at 586–598 nm, respectively, which can be attributed to the π*–π transitions.

The Zn complexes exhibit intense luminescence upon excitation, with a wavelength of 380–400 nm. The emission spectrum of the solid samples has the appearance of a broad band, with maxima in the range from 475 to 506 nm and the same maxima in a solution ranging from 473 to 504 nm. It is noteworthy that the emission bands of the complexes are hypsochromically shifted relative to the emission bands of the corresponding ligands. (Figure 7). In addition, it is noteworthy that the introduction of a fluorine atom into the *meta* position leads to a bathochromic shift of the emission band, while the introduction into the *ortho* or *para* position, with respect to the azomethine group, leads to a hypsochromic shift which is well seen for the spectra of complexes in solutions in which intermolecular effects are absent (Figure 8). This is consistent with the combined effect of the electron-donor azomethine group and electron-acceptor fluorine atom, leading to an increase in the energy difference of the frontier molecular orbitals when fluorine atoms are introduced into *ortho/para* positions and the opposite effect in the case of metasubstitution.

Solid-state complexes **2a**–**h** exhibit blue or greenish-blue luminescence, which makes them promising materials for OLED devices. The chromaticity coordinates (CIE 1932) are presented in Table 6. The PL spectra of zinc complexes recorded in dichloromethane solution show fluorescence of moderate intensity. The emission maxima are in the region of 473–511 nm.

The luminescence QYs of the complexes in a CH_2_Cl_2_ solution were measured by the relative method using quinine sulfate (QY = 0.546) as a reference. Remarkably, the complexes exhibit low QYs (<10%) in solution but much more intense emission features in the solid state. This phenomenon may be related to aggregation-induced emission enhancement (AIEE), in which the intramolecular rotation of the flexible structural element is limited in the solid state, resulting in higher PL efficiency.

The luminescence decay profiles of zinc complexes were measured at optimal excitation wavelengths. The detailed data are summarized in Table 6. For both solid samples and solutions, the emission decays can only be approximated by monoexponential functions. The general trend is that the luminescence lifetimes in the solid state (τ = 5.8–9.1 ns) are longer than in solution (τ = 3.0–3.8 ns), which may be due to their less polar nature in the solid state.

### 3.6. OLED—Performances

To evaluate and compare the EL properties of **2a**–**h** complexes, we used them as emitting materials in the fabrication of OLED devices by vacuum deposition. The EL cells were constructed as follows: ITO/PEDOT:PSS/complexes/TPBi/(Ca|Al), in which complexes **2a**–**h** act as emitters, PEDOT:PSS (poly(3,4-ethylenedioxythiophene) polystyrene sulfonate) is a hole injector, TPBI 1,3,5-tris(*N*-phenylbenzimidazol-2-yl)benzene is an electron transporter, and Ca/Al alloy served as the cathode

For all eight cells obtained, EL was detected at voltages higher than 3.4–4.2 V. The EL was perceived by the eye as blue or blue–green. EL spectra (Figure 9), in general by the shape and position of the maximum coincide with the spectra of PL, which indicates that the complexes act as emitters and emission signals from the electroplex or excimer/exciplex in these devices at electroexcitation, are not detected. The applied voltage has no effect on the position of the peak of the emitted light. From Table 7, showing the performance of the OLEDs, it is clear that the best values of performance and brightness are demonstrated by complexes **2c**,**f**,**g**. The luminescence intensity of devices is directly proportional to the increase in voltage. Thus, at voltage values of 11–12 V, the brightness reaches more than 1100–6300 cd/m^2^. A further increase in the bias voltage leads to electrical breakdown, leading to rapid cell destruction. Complex **2g**, with a 3,5-difluorophenyl moiety, shows the highest device efficiency, with a current efficiency of 19.7 cd A^−1^ and an EQE_max_ of 4.8%, which can be attributed to its high PLQY, efficient energy transfer between the layers of the device, and excellent quality of the thin film. It should be noted that the EQE value at 100 Cd more adequately characterizes the OLED performance in practice and is not much lower than the maximum value.

In order to improve the EL parameters, devices containing complex **2g** doped (weight doping concentrations of 5 wt %) with the host matrix with different host molecules (mCP, NPB, TAPC, and CBP) were fabricated and examined. CBP-, NPB-, and TAPC-based devices showed worse performance compared to the underdoped systems due to unfavorable energy transfer from host to guest, as evidenced by the appearance of additional bands in the EL spectrum associated with matrix sobwenn emission. At the same time, the performance using mCP was better, both in terms of brightness and efficiency (Table 8).

### 3.7. Biological Activity

The obtained azomethines **1a**–**h** and zinc(II) **2a**–**h** complexes were tested for their protistocidal, fungistatic, and antibacterial activities. The results of the tests are summarized in Table 9.

It was found that neither ligands **1a**–**h** nor zinc complexes **2a**–**h** have fungistatic activity against *Penicillium italicum*. Among azomethines, only compounds **1a**,**b**,**e** had antibacterial activity against *Staphylococcus aureus*. Their activity was 2–2.5 times weaker than the reference drug furazolidone in the cases of **1a**,**b** containing one fluorine atom in the amine part, and 2.8 times in the case of **1e** containing two fluorine atoms. Against *Escherichia coli*, **1a**–**d** and **1f**,**g** were active. The activity of **1a**,**d** was 1.8 times weaker than that of furazolidone, the activity of **1b**,**f**,**g** was 2.25 times weaker, and that of **1c** was 2.6 times weaker. Azomethines **1c**,**d**,**f**–**h** and **1e**,**h** did not show antibacterial activity against *Escherichia coli* and against *Staphylococcus aureus*, respectively.

Complexes **2a**,**d**,**e**,**g**, containing one or two fluorine atoms in the amine part of the ligand, showed bacteriostatic activity against *Staphylococcus aureus* that is more than two times lower than furazolidone. The same level of antibacterial activity relative to furazolidone is exhibited by complexes **2a**,**c**,**e**–**g** against *Escherichia coli*. Complexes **2b**,**c**,**f**,**h** were not active against *Staphylococcus aureus* and **2b**,**d**,**h** against *Escherichia coli*. The azomethines **1a**,**c**,**f**,**h** and their complexes had almost the same antibacterial activity. While **1b** had moderate bacteriostatic activity, its complex **2b** is inactive. The activity of **1e** and **2e** is almost similar against *Staphylococcus aureus*. Ligands **1d**,**g** were not active against *Staphylococcus aureus*, while complexes **2d**,**g** showed medium activity.

In the study of protistocidal properties (Table 9), it was found that azomethines **1a**,**b**, containing one fluorine atom, had high activity. The activity of **1b** was the same as that of the reference drug chloroquine and eight times stronger than the activity of the second reference drug, toltrazuril, while the activity of **1a** was four times higher than that of toltrazuril but twice as weak as chloroquine. The compound **1h**, containing three fluorine atoms in the amine part of azomethine, showed the same activity as toltrazuril but was eight times weaker than chloroquine. Azomethines **1c**,**e**,**g** showed weak protistocidal activity, which was 2–8 times weaker than that of toltrazuril. The compounds **1d**,**f** did not show protistocidal activity.

Among the complexes **2a**–**h**, the most active against *Colpoda steinii* is **2f**, which contains fluorine atoms in the three and four positions of the amine part of the ligand, the activity of which is 4.1 times stronger than that of chloroquine and 33 times higher than that of toltrazuril. The activity of **2b**,**g**,**h** was weaker than that of toltrazuril by two, eight, or four times, respectively. The protistocidal activity of **2a**–**h** decreased compared to **1a**–**h**, except for **2f**, which contained fluorine atoms in the three and four positions of the amine part of the ligand. The compound **1f** had no protistocidal activity, **2f** exhibited 4.1 times stronger activity than chloroquine and 33 times stronger than toltrazuril. At the same time, complexes **2a**,**c**–**e**, containing fluorine atoms in the *ortho*-position of the amine part of the ligand, showed no protistocidal activity.

When comparing the biological activity of chloro- [10,11] and fluoro-substituted *N*-[2-(phenyliminomethyl)phenyl]-4-methylbenzenesulfamides, it was found that azomethine **1b** has antibacterial activity against *Staphylococcus aureus* (50% of the activity of furazolidone), whereas its 4-chloro-substituted analog does not have such activity. Azomethine with 3,4-difluoroaniline **1f** has no activity, while its 3,4-dichlorosubstituted analog has an activity that is 45% of that of furazolidone. Against *Escherichia coli*, 4-fluorosubstituted **1b** and its 4-chlorosubstituted analog showed equal activity, 3,4-difluorosubstituted **1f** had activity amounting to 45% of that of furazolidone, and azomethine with 3,4-dichloroaniline had no activity. In the case of the protistocidal activity in azomethines, the substitution of chlorine in the four positions of the aniline moiety with fluorine leads to a strong increase in activity, which is comparable to that of chloroquine.

Among the zinc complexes of chloro- and fluoro-substituted *N*-[2-(phenyliminomethyl)phenyl]-4-methylbenzenesulfamide, only the 4-chloro-substituted complex was found to have antimicrobial properties *against Staphylococcus aureus* [10], and the 4-fluoro-substituted **2f** was the most active against *Escherichia coli*. In the case of protistocidal activity, the substitution of chlorine atoms in the 4-chloro- and 3,4-dichloro-substituted complexes, which do not possess this activity [10], with fluorine atoms leads to the appearance of strong protistocidal activity. So, 4-fluoro-substituted **2b** has an activity that is 50% of that of toltrazuril, and 3,4-difluoro-substituted **2f** is 4.1 times stronger than chloroquine and 33 times stronger than toltrazuril.

## 4. Conclusions

The eight new azomethine compounds of *N*-[2-(phenyliminomethyl)phenyl]-4-methylbenzenesulfonamides derivatives with fluorine-substituted anilines and their zinc(II) complexes were obtained. The composition and structure of the obtained compounds were determined by IR, 1H NMR spectroscopy, and elemental analysis. The structures of two azomethines *N*-[2-[[(*E*)-(2,5-difluorophenyl)iminomethyl]phenyl]phenyl]-4-methyl-benzenesulfonamide and *N*-[2-[[[(*E*)-(3,4-difluorophenyl)iminomethyl]phenyl]phenyl]-4-methyl-benzenesulfonamide and their zinc(II) complexes, as well as Bis[*N*-(*p*-tolylsulfonyl)-2-[(*E*)-(2,4,6-trifluorophenyl)iminomethyl]anilino]zinc(II) were determined by single-crystal X-ray diffraction. Azomethines in the solid state have PL-band maxima in the region λPl 567–589 nm, with PL quantum yields from 10.3 to 43.9%, which are 4–16 times higher than the corresponding quantum yields for unsubstituted azomethines. In the PL spectra of zinc(II) complexes, the maxima of the PL bands undergo a hypsochromic shift to 476–506 nm compared to free ligands, and their quantum yields range from 22.3 to 42.2%. The obtained quantum yields for zinc complexes with fluorine-substituted ligands are 2–4 times higher compared to the quantum yield for the zinc complex with unsubstituted ligands. OLEDs were made using zinc complexes as emissive layers, for which the maximum brightness was from 950 to 8120 cd/m^2^ with a current efficiency of 6.9 to 21.1 cd/A. Obtained results are comparable and even higher in some cases with the same class of luminophores [28,29,30,31,32]. The obtained azomethines and zinc(II) complexes were tested for their protistocidal, fungistatic, and antibacterial activities. It was shown that the replacement of chlorine atoms in 3,4-dihalophenyl-substituted complexes with fluorine atoms leads to the appearance of strong protistocidal activity in the complexes.

## Data Availability

The data presented in this article are openly available.

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
