# Peer review of "Zinc Complexes of Fluorosubstituted N-[2-(Phenyliminomethyl)phenyl]-4-methylbenzenesulfamides: Synthesis, Structure, Luminescent Properties, and Biological Activity"

_materials, 2024, doi:10.3390/ma17020438_

Round 1

Reviewer 1 Report

Comments and Suggestions for Authors

The synthesis, PL and EL properties of azomethine compounds of 2-hydroxybenzaldehydes and 2-N-tosylaminobenzaldehydes with fluorine-substituted aromatic amines in different positions and the corresponding zinc(II) complexes were conducted in this work. The compounds were comprehensively characterized by spectroscopic, X-ray crystallography and photophysical studies. The OLEDs based on zinc(II) complexes were investigated, giving some interesting results which could be published after appropriate revisions by following revision suggestions.

11.  The purpose of this research could be elucidated more clearly in the introduction section.  

22.  The conclusion should be rewritten. Generally, the conclusion should contain the critical finding and valuable information from the research work.

33.  How were the single crystals of zinc (II) complexes grew ? The crystal growth condition should be given.

44.  Excitation spectra should be provided and given in Figure 7.

55.      “In addition, it is noteworthy that the introduction of a fluorine atom into the meta position leads to a bathochromic shift of the emission band, while the introduction into the ortho position leads to a hypsochromic shift (Fig. 8). “ Why ?  Reasonable interpretation must be provided in revision version. Particularly, the emission spectral shifts in CH2Cl2 solutions could be elucidated perfectly based on electronic effects of the substituents for zinc(II) complexes 2a-2h.

66.  The luminescent data of doping films could be put in Table 6.

47.  The turn-on voltage and powder efficiency of various OLEDs could be provided in Tables 6 and 7.

88.  Representative V−J−L characteristics and CE/EQE vs luminance for the OLEDs must be provided in main text.

Comments on the Quality of English Language

The manuscript should be carefully edited.  

Author Response

We thank the Referees for their interest in our work and for helpful comments and constructive suggestions that will greatly improve the quality of this manuscript. As indicated below, we have checked all the general and specific comments provided by the Referees and have made necessary changes according to their indications (the reviewer’s comments are in italics).

Reviewers' comments:

Reviewer #1: The synthesis, PL and EL properties of azomethine compounds of 2-hydroxybenzaldehydes and 2-N-tosylaminobenzaldehydes with fluorine-substituted aromatic amines in different positions and the corresponding zinc(II) complexes were conducted in this work. The compounds were comprehensively characterized by spectroscopic, X-ray crystallography and photophysical studies. The OLEDs based on zinc(II) complexes were investigated, giving some interesting results which could be published after appropriate revisions by following revision suggestions.

Reviewer #1: The purpose of this research could be elucidated more clearly in the introduction section.

A: The introduction section has been revised to specify the aims and objectives of the research

Reviewer #1: The conclusion should be rewritten. Generally, the conclusion should contain the critical finding and valuable information from the research work.

A: The conclusion to the article has been completely rewritten.

Reviewer #1: How were the single crystals of zinc(II) complexes grew? The crystal growth condition should be given.

A: The conditions for crystal growth have been added to the text of the manuscript. The single crystals of zinc(II) complexes were grown by slow evaporation at room temperature of their solutions in a mixture of methylene chloride and methanol (1:2).

Reviewer #1: Excitation spectra should be provided and given in Figure 7.

A: Excitation spectra are added to Figure 7.

Reviewer #1: In addition, it is noteworthy that the introduction of a fluorine atom into the meta-position leads to a bathochromic shift of the emission band, while the introduction into the ortho-position leads to a hypsochromic shift (Fig. 8). “ Why ?  Reasonable interpretation must be provided in revision version. Particularly, the emission spectral shifts in CH2Cl2 solutions could be elucidated perfectly based on electronic effects of the substituents for zinc(II) complexes 2a-2h.

A: A brief comment has been inserted in the appropriate section

Reviewer #1: The luminescent data of doping films could be put in Table 6.

A: Unfortunately photoluminescence spectra for thin films were not recorded. Therefore, it is not possible to describe them. Meanwhile, it is known from many references that the spectral characteristics of complexes in the solid state and films correlate with each other, so we do not expect any significant changes

Reviewer #1: The turn-on voltage and powder efficiency of various OLEDs could be provided in Tables 6 and 7.

A: Turn-on voltage are inserted to Table 7

Reviewer #1: Representative V−J−L characteristics and CE/EQE vs luminance for the OLEDs must be provided in main text.

A: The relevant figures have been added to the text of the article

Reviewer 2 Report

Comments and Suggestions for Authors

The authors present an experimental study on a group of Zn complexes with interesting luminescent properties, with possible application in OLED technology. The biological activity was also assessed. The research was performed correctly and the results are described properly. However, I have some minor remarks.

1.     Please explain what was the aim to study the bioactivity of the new compounds together with the possible application in OLED technology? Is there any reason to combine these studies? Please justify.

2.     There are a lot of international research groups which publish every year very interesting results in similar research area. However, there are missing the references to the most recent papers reporting current advances in this field. Instead, there are some references older than 50 years. If such references are necessary please try to justify it more clearly. In my opinion the authors do not compare their results sufficiently with the current state of knowledge. I have also an impression that the self-citation rate is too high. Please correct it.

Author Response

We thank the Referees for their interest in our work and for helpful comments and constructive suggestions that will greatly improve the quality of this manuscript. As indicated below, we have checked all the general and specific comments provided by the Referees and have made necessary changes according to their indications (the reviewer’s comments are in italics).

Reviewer #2: The authors present an experimental study on a group of Zn complexes with interesting luminescent properties, with possible application in OLED technology. The biological activity was also assessed. The research was performed correctly and the results are described properly. However, I have some minor remarks.

  1. Please explain what was the aim to study the bioactivity of the new compounds together with the possible application in OLED technology? Is there any reason to combine these studies? Please justify.

A: The reviewer is quite right to note the unification of seemingly different areas of research related to materials science and biological activity. We generally agree with this statement. However, the wide range of properties of our synthesized compounds (both electroluminescent and antibacterial properties) prompted us to acquaint the readers with all their properties. Meanwhile, taking into account the profile of the journal, we have moved part of the material related to biological activity to the supplementary section.

  1. There are a lot of international research groups which publish every year very interesting results in similar research area. However, there are missing the references to the most recent papers reporting current advances in this field. Instead, there are some references older than 50 years. If such references are necessary please try to justify it more clearly. In my opinion the authors do not compare their results sufficiently with the current state of knowledge. I have also an impression that the self-citation rate is too high. Please correct it.

A: We are grateful for this comment. We have added to the list of references the most important works in the field of electroluminescence of zinc complexes performed during the last 5 years. Self-citation in this case is necessary to indicate the connection with the previous works with azomethine derivatives of 2-N-tosylaminobenzaldehyde, which are carried out by our group.

Reviewer 3 Report

Comments and Suggestions for Authors

In this manuscript the authors present a synthesis of multiple zinc(II) complexes with azomethine ligands and study their structure, photophysical behaviour and bioactivityThe results are well-explained by scientific evidence, methods and experimental part provides enough details to ensure the reproducibility and further development in this research direction by their peers in the community. This manuscript would be of interest the synthetic, photophysical and pharmaceutical scientists.  The manuscript requires a proof-read due to some typos and unclear statements. Biological activity is outside of my expertise therefore my comments are related to the synthesis, characterisation and OLED devices.  

Overall, I suggest acceptance of this manuscript after minor revision and corrections to improve the quality of the manuscript, please, see below.  

Authors should develop a consistent introduction with some extra details to prepare the reader for their choice of the synthetic targets, electroluminescence study and rationale to explore bioactivity. In particular:  

Lines 30-38. Disagree. There are numerous blue phosphors. The authors should clearly state the problem in the OLED industry. It’s not the number of the blue emitters but the operational stability of those under electrical excitation that requires an enhancement to design a next generation of the energy efficient OLED and white light sources.  

Lines 41-44. Please provide a reference for the line “ The replacement of one or more C-H bonds by C-Cl and C-F bonds in the amine or 42 aldehyde moieties of azomethines also leads to an increase in the PL QYs of the coordination compounds due to the quenching of vibrations.”  

Lines 39-53. Please, be specific what metal complexes do you mean? The effect would be very different across the transition metals. Explain your choice on translating azomethine ligand system from lanthanoid to the Zinc. Please, avoid ambiguity since TADF transition metal complexes and classical organic TADF emitters suffer from the introduction of the heavier halides as phosphorescence becomes more allowed. The reader may be confused here.  

It might be helpful to note that installation of the electron withdrawing fluorine atoms help to stabilize HOMO to widen the energy gap of the materials thus realizing the blue emitter. This will help you to link two paragraphs together. 

Just make the introduction consistent and a linked story. 

Line 120. Change “Creation” to “Fabrication of the OLED stack” 

Scheme 1, Line 357. Chemical error for 2a-h. The digit 2 after dash should refer only to the ligand and not to the ligand and the zinc! At the moment the scheme suggests the synthesis of the di zinc complexes with a Zn-Zn bond. Please correct to represent the mononuclear zinc complex ZnL2.  

Lines 370-372. Correct confusing statement. “a shift to 370 the long wavelength (low frequency) region at 8-28 cm-1, and the bands of νasSO371 at 38 - 47 cm-1 and νsSOat 22 - 28 cm-1.” Please use correct wording throughout the manuscript for the IR, i.e. it’s not “at 8-28 cm-1” but “by 8-28 cm-1” and so one. Authors discuss the shift and not a position of the vibrations in the range below 100 cm-1. 

Line 381 Figure caption 1. Add “intramolecular hydrogen bonds” to be specific. 

Lines 402-411. Authors invoke words “weaker” and “stronger” hydrogen bonds thus suggesting the comparison at least with the sum of the Van der Waals radii of the interacting atoms to support the statements. Please label on Figure 2 the key atoms involved into the discussion of the contacts and key bond lengths in this paragraph to enhance the graphics and text correlation.  

Line 416. Table 2. Why authors omit errors for the C-H…O hydrogen bond lengths? Please, add and correct text accordingly. It could be helpful to add packing and stacking diagram to the ESI. Figure 2 should show cell axis.  

Endnote for Table 2. Symmetry equivalent: a-1+x,y,z; b-x,1-y,1-z. Be consistent within one manuscript: either symmetry code (Figure 4-6) or equivalent (Table 2)? 

Line 556-559. “It is noteworthy that the emission bands of the complexes are hypsochromically shifted relative to the emission bands of the corresponding ligands, which may be due to tautomeric transformations during coordination of azomethines to the zinc cation (Fig. 7).” Be careful by suggesting this, azomethines and many other ligands capable of tautomeric forms emit light via a charge transfer type excited state intramolecular proton transfer (ESIPT). One would have a change in emission mechanism upon coordination of the zinc from the ESIPT to ligand-based fluorescence (the case of authors) or rarer cases of the zinc-based TADF or phosphorescent materials. 

Line 593. “in the fabrication of OLED displays” correct to “in the fabrication of OLED devices”  

Line 603. “exiplex” correct typo to “exciplex” 

Page 19. Tables 7 and 8. EQE values should be noted for the practical brightness at 100 nits not only for the EQEmax value which is not reproducible in majority of cases. Current-voltage, current-brightness, current-EQE graphs should be shown in the supporting information which would also facilitate the reader with the information on efficiency roll-off for the OLED pixels.  

Table 8. Authors may provide a suggestion to explain efficiency 5.1% of the OLED device that exceeds the theoretical max EQE of 5% for the fluorescent materials. 

Author Response

We thank the Referees for their interest in our work and for helpful comments and constructive suggestions that will greatly improve the quality of this manuscript. As indicated below, we have checked all the general and specific comments provided by the Referees and have made necessary changes according to their indications (the reviewer’s comments are in italics).

Reviewer #2: In this manuscript the authors present a synthesis of multiple zinc(II) complexes with azomethine ligands and study their structure, photophysical behaviour and bioactivity. The results are well-explained by scientific evidence, methods and experimental part provides enough details to ensure the reproducibility and further development in this research direction by their peers in the community. This manuscript would be of interest the synthetic, photophysical and pharmaceutical scientists.  The manuscript requires a proof-read due to some typos and unclear statements. Biological activity is outside of my expertise therefore my comments are related to the synthesis, characterisation and OLED devices. Overall, I suggest acceptance of this manuscript after minor revision and corrections to improve the quality of the manuscript, please, see below. Authors should develop a consistent introduction with some extra details to prepare the reader for their choice of the synthetic targets, electroluminescence study and rationale to explore bioactivity.

A: We thank the reviewer #2 for careful reading of the manuscript and helpful comments. In accordance with the referee's comments, we made the following corrections in the text of the manuscript:

         Reviewer #2: Lines 30-38. Disagree. There are numerous blue phosphors. The authors should clearly state the problem in the OLED industry. It’s not the number of the blue emitters but the operational stability of those under electrical excitation that requires an enhancement to design a next generation of the energy efficient OLED and white light sources.

A: The introduction chapter has been completely rewritten

Reviewer #2: Lines 41-44. Please provide a reference for the line “ The replacement of one or more C-H bonds by C-Cl and C-F bonds in the amine or 42 aldehyde moieties of azomethines also leads to an increase in the PL QYs of the coordination compounds due to the quenching of vibrations.”

A: Similar conclusions were made in [7-9].

  1. Hasegawa, Y.; Wada, Y.; Yanagida, S.; Strategies for the design of luminescent lanthanide(III) complexes and their photonic applications. J. Photochem. Photobio. C: Photochem. Rev. 2004, 5(3), 183-202. [https://doi.org/10.1016/j.jphotochemrev.2004.10.003] 
  2. Artizzu, F.; Mercuri, M.L.; Serpe, A.; Deplano, P. NIR-emissive erbium–quinolinolate complexes. Coord. Chem. Rev. 2011, 255, 2514–2529, [https://doi.org/10.1016/j.ccr.2011.01.013]
  3. Zheng, Y.; Motevalli, M.; Tan, R.H.C.; Abrahams, I.; Gillin, W.P.; Wyatt, P.B. Near IR luminescent rare earth 3,4,5,6-tetrafluoro-2-nitrophenoxide complexes: synthesis, X-ray crystallography and spectroscopy. Polyhedron 2008, 27, 1503–1510. [https://doi.org/10.1016/j.poly.2008.01.022]

Reviewer #2: Lines 39-53. Please, be specific what metal complexes do you mean? The effect would be very different across the transition metals. Explain your choice on translating azomethine ligand system from lanthanoid to the Zinc. Please, avoid ambiguity since TADF transition metal complexes and classical organic TADF emitters suffer from the introduction of the heavier halides as phosphorescence becomes more allowed. The reader may be confused here. It might be helpful to note that installation of the electron withdrawing fluorine atoms help to stabilize HOMO to widen the energy gap of the materials thus realizing the blue emitter. This will help you to link two paragraphs together. Just make the introduction consistent and a linked story. 

A: The authors explained their choice of synthesis of fluorine-substituted azomethine ligand systems and zinc complexes based on them.

Reviewer #2: Line 120. Change “Creation” to “Fabrication of the OLED stack”

A: The suggested correction has been made.

Reviewer #2: Scheme 1, Line 357. Chemical error for 2a-h. The digit 2 after dash should refer only to the ligand and not to the ligand and the zinc! At the moment the scheme suggests the synthesis of the di zinc complexes with a Zn-Zn bond. Please correct to represent the mononuclear zinc complex ZnL2.

A: The suggested correction has been made. Scheme 1 has been modified in accordance with the reviewer's comments.

Reviewer #2: Lines 370-372. Correct confusing statement. “a shift to 370 the long wavelength (low frequency) region at 8-28 cm-1, and the bands of νasSO2 371 at 38 - 47 cm-1 and νsSO2 at 22 - 28 cm-1.” Please use correct wording throughout the manuscript for the IR, i.e. it’s not “at 8-28 cm-1” but “by 8-28 cm-1” and so one. Authors discuss the shift and not a position of the vibrations in the range below 100 cm-1

A: The suggested correction has been made.

Reviewer #2: Line 381 Figure caption 1. Add “intramolecular hydrogen bonds” to be specific. 

A: The suggested correction has been made.

Reviewer #2: Lines 402-411. Authors invoke words “weaker” and “stronger” hydrogen bonds thus suggesting the comparison at least with the sum of the Van der Waals radii of the interacting atoms to support the statements. Please label on Figure 2 the key atoms involved into the discussion of the contacts and key bond lengths in this paragraph to enhance the graphics and text correlation.  

A: Authors replaced the words “weaker” and “stronger” with the expression “intramolecular hydrogen bonds”. The characteristics of these H-bonds are listed in Table 2. Figure 2 shows еhe antiparallel π-stacking interactions in compound 1f. We consider it unnecessary to show packaging for 1в, since these data are given in table 2.

Reviewer #2: Line 416. Table 2. Why authors omit errors for the C-H…O hydrogen bond lengths? Please, add and correct text accordingly. It could be helpful to add packing and stacking diagram to the ESI. Figure 2 should show cell axis. Endnote for Table 2. Symmetry equivalent: a-1+x,y,z; b-x,1-y,1-z. Be consistent within one manuscript: either symmetry code (Figure 4-6) or equivalent (Table 2)?

A: The errors for the C-H…O hydrogen bond lengths in Table 2 were added. The cell axis is shown in Figure 2. The phrase “symmetry code” has been replaced “Symmetry equivalent”.

Reviewer #2: Line 556-559. “It is noteworthy that the emission bands of the complexes are hypsochromically shifted relative to the emission bands of the corresponding ligands, which may be due to tautomeric transformations during coordination of azomethines to the zinc cation (Fig. 7).” Be careful by suggesting this, azomethines and many other ligands capable of tautomeric forms emit light via a charge transfer type excited state intramolecular proton transfer (ESIPT). One would have a change in emission mechanism upon coordination of the zinc from the ESIPT to ligand-based fluorescence (the case of authors) or rarer cases of the zinc-based TADF or phosphorescent materials. 

A: We agree with the reviewer that this statement is not supported by data, so we decided to omit the conclusion about the cause of the emission band shift

Reviewer #2: Line 593. “in the fabrication of OLED displays” correct to “in the fabrication of OLED devices”  

A: The suggested correction has been made.

Reviewer #2: Line 603. “exiplex” correct typo to “exciplex” 

A: The suggested correction has been made.

Reviewer #2:  Page 19. Tables 7 and 8. EQE values should be noted for the practical brightness at 100 nits not only for the EQEmax value which is not reproducible in majority of cases. Current-voltage, current-brightness, current-EQE graphs should be shown in the supporting information which would also facilitate the reader with the information on efficiency roll-off for the OLED pixels.  

A: We completely agree with this comment. Therefore, appropriate comments have been added to the text and to Tables 7 and 8 indicating the efficacy at 100 Cd/m2

Reviewer #2: Table 8. Authors may provide a suggestion to explain efficiency 5.1% of the OLED device that exceeds the theoretical max EQE of 5% for the fluorescent materials. 

A: The reviewer is correct in pointing out this inconsistency. The available data do not allow us to give an unambiguous answer, but we believe that since this value is obtained at low voltage, perhaps under these conditions there is a more complex mechanism of electroluminescence leading to a large value of efficiency.

Reviewer 4 Report

Comments and Suggestions for Authors

In this manuscript submitted to Materials, the authors detail the synthesis of eight azomethine compounds of the type "(E)-4-methyl-N-(2-((phenylimino)methyl)phenyl)benzenesulfonamide," incorporating fluorine atoms at various positions in the aromatic ring of the (phenylimino)methyl moiety. The subsequent synthesis involves the preparation of eight corresponding bis(azomethine)Zinc(II) complexes. Comprehensive characterization was performed using spectroscopic, spectrometric methods, and synchrotron radiation crystallography, covering two azomethines and three zinc complexes.

The authors present findings indicating that the introduction and increment of fluorine atoms in the ligand structure enhance luminescence quantum yields, as well as performance and brightness in electroluminescence cells, in comparison to their previously studied chlorine-substituted analogues.

From the review of the manuscript, the following corrections must be made:

Scheme 1, Figure 2a-h: Corrections are needed to accurately represent a complex with two ligands and a metal: (azomethine)2Zn, rather than a dinuclear complex (azomethine)Zn/2. Please make the necessary adjustments.

Page 21, Row 710: Replace "protiscidal activity" with the correct term "protistocidal activity."

Crystallographic Structural Resolutions: Four out of the five structural resolutions (1d, 1f, 2d, and 2h) are flawless. However, for complex 2f (CCDC 2299397) it exhibits defects requiring correction. The reviewer has re-examined and refined this structure using the ".res" and ".hkl" files from the deposited ".cif" file. The issues include modelling errors related to positional disorder of fluorine and hydrogen in the (phenylimino)methyl moiety. Specifically, one azomethine ligand, modeled correctly, inadvertently involved a crystallizing methanol molecule. This solvent molecule must be refined without disorder. Additionally, errors in the atoms associated with fluorine atoms, resulting from disorder-related restrictions on C-F bonds using the shelxl SADI command, need correction. Finally, the UNIT entry associated with the SFAC must be fixed. Generate a new CIF reflecting these corrections and update it in the CCDC.

A snippet of the new checkCIF obtained is provided below as an example:
--------------------------------------------

Datablock: m206_mer

Bond precision:          C-C = 0.0031 A            Wavelength=0.74500

Cell:     a=8.4800(17)  b=11.858(2)    c=19.371(4)

alpha=81.98(3)           beta=79.75(3) gamma=73.71(3)

Temperature: 100 K              

Calculated       Reported

Volume           1831.5(7)        1831.5(7)

Space group    P -1     P -1

Hall group       -P 1     -P 1

Moiety formula          C40 H30 F4 N4 O4 S2 Zn, C H4 O       C40 H30 F4 N4 O4 S2 Zn, C H4 O

Sum formula   C41 H34 F4 N4 O5 S2 Zn        C41 H34 F4 N4 O5 S2 Zn

Mr       868.23 868.21

Dx,g cm-3       1.574   1.574

Z          2          2

Mu (mm-1)     0.972   0.949

F000    892.0   892.0

F000'   893.54

h,k,lmax          11,16,26          11,16,26

Nref    10158  9809

Tmin,Tmax     0.945,0.972     0.866,1.000

Tmin'  0.867  

Correction method= # Reported T Limits: Tmin=0.866 Tmax=1.000 AbsCorr = EMPIRICAL          

Data completeness= 0.966    Theta(max)= 31.007

R(reflections)= 0.0448( 9043)            wR2(reflections)= 0.1295( 9809)

S = 1.037         Npar= 528

The following ALERTS were generated. Each ALERT has the format

       test-name_ALERT_alert-type_alert-level.

Click on the hyperlinks for more details of the test.

Alert level C

PLAT911_ALERT_3_C Missing FCF Refl Between Thmin & STh/L=    0.600        265 Report

 ------------------------------------------------

Please address these corrections and ensure that the revised manuscript accurately reflects the corrections provided in this review.

Author Response

Reviewer #4: In this manuscript submitted to Materials, the authors detail the synthesis of eight azomethine compounds of the type "(E)-4-methyl-N-(2-((phenylimino)methyl)phenyl)benzenesulfonamide," incorporating fluorine atoms at various positions in the aromatic ring of the (phenylimino)methyl moiety. The subsequent synthesis involves the preparation of eight corresponding bis(azomethine)Zinc(II) complexes. Comprehensive characterization was performed using spectroscopic, spectrometric methods, and synchrotron radiation crystallography, covering two azomethines and three zinc complexes. The authors present findings indicating that the introduction and increment of fluorine atoms in the ligand structure enhance luminescence quantum yields, as well as performance and brightness in electroluminescence cells, in comparison to their previously studied chlorine-substituted analogues.

Reviewer #4: Scheme 1, Figure 2a-h: Corrections are needed to accurately represent a complex with two ligands and a metal: (azomethine)2Zn, rather than a dinuclear complex (azomethine)Zn/2. Please make the necessary adjustments.

A: The suggested correction has been made. Scheme 1 has been modified in accordance with the reviewer's comments.

Reviewer #4: Page 21, Row 710: Replace "protiscidal activity" with the correct term "protistocidal activity."

A: The suggested correction has been made.

Reviewer #4: Crystallographic Structural Resolutions: Four out of the five structural resolutions (1d, 1f, 2d, and 2h) are flawless. However, for complex 2f (CCDC 2299397) it exhibits defects requiring correction. The reviewer has re-examined and refined this structure using the ".res" and ".hkl" files from the deposited ".cif" file. The issues include modelling errors related to positional disorder of fluorine and hydrogen in the (phenylimino)methyl moiety. Specifically, one azomethine ligand, modeled correctly, inadvertently involved a crystallizing methanol molecule. This solvent molecule must be refined without disorder. Additionally, errors in the atoms associated with fluorine atoms, resulting from disorder-related restrictions on C-F bonds using the shelxl SADI command, need correction. Finally, the UNIT entry associated with the SFAC must be fixed. Generate a new CIF reflecting these corrections and update it in the CCDC.
A snippet of the new checkCIF obtained is provided below as an example:
--------------------------------------------

Datablock: m206_mer

Bond precision:          C-C = 0.0031 A            Wavelength=0.74500

Cell:     a=8.4800(17)  b=11.858(2)    c=19.371(4)

alpha=81.98(3)           beta=79.75(3) gamma=73.71(3)

Temperature: 100 K              

Calculated       Reported

Volume           1831.5(7)        1831.5(7)

Space group    P -1     P -1

Hall group       -P 1     -P 1

Moiety formula          C40 H30 F4 N4 O4 S2 Zn, C H4 O       C40 H30 F4 N4 O4 S2 Zn, C H4 O

Sum formula   C41 H34 F4 N4 O5 S2 Zn        C41 H34 F4 N4 O5 S2 Zn

Mr       868.23 868.21

Dx,g cm-3       1.574   1.574

Z          2          2

Mu (mm-1)     0.972   0.949

F000    892.0   892.0

F000'   893.54

h,k,lmax          11,16,26          11,16,26

Nref    10158  9809

Tmin,Tmax     0.945,0.972     0.866,1.000

Tmin'  0.867  

Correction method= # Reported T Limits: Tmin=0.866 Tmax=1.000 AbsCorr = EMPIRICAL          

Data completeness= 0.966    Theta(max)= 31.007

R(reflections)= 0.0448( 9043)            wR2(reflections)= 0.1295( 9809)

S = 1.037         Npar= 528

The following ALERTS were generated. Each ALERT has the format

       test-name_ALERT_alert-type_alert-level.

Click on the hyperlinks for more details of the test.

Alert level C

PLAT911_ALERT_3_C Missing FCF Refl Between Thmin & STh/L=    0.600        265 Report

 ------------------------------------------------

 Please address these corrections and ensure that the revised manuscript accurately reflects the corrections provided in this review.

A: The structure 2f has been carefully revised. SADI restrictions were cleared with subsequent improvement of R work. Fluorine of the (phenyl)iminomethyl fragments has two conformational positions with different occupancies, which were defined separately. In the case of conformation "part 2", F1 is too close to the O5 atom of methanol (2.005 Å). Thus, the methanol molecule is present in the structure of 2f only in the case of conformation “part 1”. Thus, the fractional chemical formula C40H30F4N4O4S2Zn, 0.632(CH4O) is a consequence of the fractional population of methanol.